# Potential Role of Circulating PD-L1^+^ Leukocytes as a Predictor of Response to Anti-PD-(L)1 Therapy in NSCLC Patients

**DOI:** 10.3390/biomedicines12050958

**Published:** 2024-04-25

**Authors:** Georgia Anguera, Maria Mulet, Carlos Zamora, Rubén Osuna-Gómez, Andrés Barba, Ivana Sullivan, Jorgina Serra-López, Elisabet Cantó, Silvia Vidal, Margarita Majem

**Affiliations:** 1Department of Medical Oncology, Hospital de la Santa Creu i Sant Pau, 08041 Barcelona, Spain; ganguera@santpau.cat (G.A.); abarba@santpau.cat (A.B.); isullivan@santpau.cat (I.S.); jserral@santpau.cat (J.S.-L.); mmajem@santpau.cat (M.M.); 2Department of Medicine, Universitat Autònoma de Barcelona, 08193 Bellaterra, Spain; 3Group of Inflammatory Diseases, Institut de Recerca Sant Pau (IR Sant Pau), 08041 Barcelona, Spain; carlosza86@gmail.com (C.Z.); rosuna@santpau.cat (R.O.-G.); ecanto@santpau.cat (E.C.); svidal@santpau.cat (S.V.); 4Department of Cell Biology, Physiology and Immunology, Universitat Autònoma de Barcelona, 08193 Bellaterra, Spain

**Keywords:** NSCLC, immunotherapy, PD-L1^+^ neutrophils, PD-L1^+^ CD14^+^ cells, PDL-1^+^ platelets

## Abstract

PD-(L)1 inhibitors are part of the treatment strategy for non-small cell lung cancer (NSCLC) although its efficacy is limited to certain patients. Our study aimed to identify patients who might benefit from anti-PD-(L)1 inhibitors by analyzing the PD-L1 expression on circulating leukocytes and its evolution during treatment. One hundred thirteen NSCLC patients, according to their radiological response after 10–12 weeks of treatment, were classified into responders, stable, and progressive disease. Percentages of circulating PD-L1^+^ leukocytes, PD-L1^+^ platelets (PLTs), and leukocyte-PLT complexes were assessed using flow cytometry, and plasma concentrations of soluble immunomodulatory factors were quantified by ELISA. Responders exhibited significantly higher pre-treatment percentages of PD-L1^+^ neutrophils, PD-L1^+^ CD14^+^ cells, and PD-L1^+^ PLTs than progressors. The percentages of these populations decreased in responders post-treatment, contrasting with stables and progressors. PLTs notably contributed to PD-L1 expression in CD14^+^ cells and neutrophils. Plasma cytokine analysis revealed baseline differences only in IL-17 concentration among groups, whereas network analyses highlighted distinct association patterns between plasma molecules and PD-L1^+^ leukocytes after 10–12 weeks of treatment. Our findings suggest that pre-treatment assessment of circulating PD-L1^+^ neutrophils, PD-L1^+^ CD14^+^ cells, and PD-L1^+^ PLTs may be helpful in identifying NSCLC patients who are potential candidates for anti-PD-(L)1 therapy.

## 1. Introduction

Immune checkpoint inhibitors (ICIs) have transformed the treatment landscape for advanced non-small cell lung cancer (NSCLC) as well as other tumors [1,2,3]. This shift in treatment paradigms has been seen since the approval of anti-PD-(L)1 therapy (nivolumab [4,5], pembrolizumab [6], and atezolizumab [7]) in the second-line setting, together with the establishment of the role of ICI in first-line therapy, both alone [8] or in combination with chemotherapy [9,10,11,12]. Unfortunately, there is still a significant proportion of patients with NSCLC that are either refractory or develop resistance to anti-PD-(L)1 therapy [13,14].

The identification of reliable biomarkers for predicting ICI response or resistance remains an unmet need. Such markers are essential for guiding treatment decisions, minimizing unnecessary toxicity, and managing economic costs. Although PD-L1 immunohistochemistry testing in tumor samples is recommended by international guidelines [15], it has several limitations, including variability in results with different immunohistochemistry clones and the requirement for enough tissue material [15], among others. Other potential biomarkers, such as tumor mutational burden and peripheral blood markers like PLT-to-lymphocyte ratio, neutrophil-to-lymphocyte ratio, and the lung immune prognostic index score, have been explored; however, their potential predictive role has not as yet been fully clarified [16,17].

In recent years, there have been significant advances in the understanding of the molecular and cellular pathogenesis of NSCLC. It is well-documented that tumor cells employ various mechanisms to evade immune surveillance, including the expression of coinhibitory molecules such as PD-L1 [10]. This molecule is not restricted to tumor cells as immune infiltrating cells can also express PD-L1. The interaction between PD-L1 and its receptor, PD-1, expressed in T cells, elicits inhibitory signals within T cells, dampening their cytotoxic activity against the tumor [14,18]. Anti-PD-(L)1 therapy disrupts this interaction, thereby unleashing the immune system’s ability to target and destroy cancer cells. Higher levels of PD-L1 expression on tumor cells are often related to a higher probability of response to anti-PD-(L)1 therapy [19]. Additionally, PD-L1 can also be detected in the bloodstream as a soluble molecule (sPD-L1), with higher concentrations observed in patients with NSCLC compared to healthy donors (HD) [20]. Increased sPD-L1 concentration after anti-PD-(L)1 therapy is associated with a lack of clinical benefit from ICI and also with a worse prognosis [21].

Historically considered primarily as blood clotting agents, PLTs have increasingly been recognized as key modulators of the immune response within the tumor microenvironment. Through both direct and indirect interactions with leukocytes, PLTs regulate several aspects of tumor-associated pathology, including tumor growth and metastasis. In a previous publication, we demonstrated that patients with NSCLC exhibit higher percentages of circulating leukocyte-PLT complexes compared to HD and, particularly, complexes of CD4^+^ T cell-PLTs and CD14^+^ cell-PLTs can be used as a predictive biomarker of the development and severity of immune-related adverse events associated to ICI therapy [22]. Recent reports also indicate that PLTs may increase PD-L1 expression in cancer conditions, potentially attenuating the antitumor immune response [23]. Hinterleitner et al. experimentally demonstrated that PD-L1 is transferred from tumor cells to PLTs preserving its immunomodulatory properties. Moreover, they also designed an algorithm considering the activation-dependent expression change in PD-L1 PLTs, demonstrating superior effectiveness in predicting ICI response compared to histologically tumor PD-L1 quantification. Patients identified with high algorithm values exhibited shorter overall survival (OS) [24].

Currently, the analysis of leukocyte and PLT phenotypes has provided valuable insights into the research of biomarkers that predict the efficacy of ICI in cancer patients. We have previously reported that patients with higher percentages of PD-L1^+^ CD14^+^ cells, PD-L1^+^ neutrophils, and PD-L1^+^ PLTs had significantly longer progression-free survival (PFS) [20]. In regard to OS, patients with higher percentages of PD-L1^+^ CD14^+^ cells and high tumor PD-L1 expression exhibited longer OS [20].

Based on all these findings, we hypothesize that baseline levels and changes in PD-L1 expression on specific leukocyte populations in the blood may influence the response to anti-PD-(L)1 treatment in advanced NSCLC patients. First, we explored the kinetics of PD-L1 expression on distinct leukocyte populations during the first 24 weeks of anti-PD-(L)1 therapy, comparing responding, stable, and progressing patients undergoing anti-PD-(L)1 treatment. Second, we determined the contribution of PLTs to PD-L1 expression in leukocytes. Finally, we constructed a correlation network to explore changes in cellular and molecular connectivity patterns among patients with different treatment responses and integrated parameters to identify the strongest predictors of response with a multinomial logistic regression.

## 2. Materials and Methods

### 2.1. Patient Cohort and Sample Collection

This prospective study included 113 patients with advanced NSCLC (IV stage) undergoing anti-PD-(L)1 treatment at a single institution (Hospital de la Santa Creu i Sant Pau, Barcelona). Whole blood samples were collected in heparinized BD Vacutainer tubes (BD, Franklin Lakes, NJ) before starting anti-PD-(L)1 therapy (t = 0) and then every 4–6 weeks until 22–24 weeks (t = 4–6 w, t = 10–12 w, t = 16–18 w, t = 22–24 w). Patient recruitment started in May 2015 and finished in September 2019, and the follow-up period ended in November 2022. The median follow-up was 58 months (39–88). All patients received at least one dose of anti-PD-(L)1 treatment. Treatment was discontinued in cases of progressive disease by Response Evaluation Criteria in Solid Tumors (RECIST) or Immune-Response Evaluation Criteria in Solid Tumors (iRECIST), loss of clinical benefit, treatment completion, or unacceptable treatment-related toxicity. All patients signed the written informed consent, and this research adhered strictly to the ethical principles outlined in the Helsinki Declaration and received approval from the Research Ethics Board of Hospital de la Santa Creu i Sant Pau. Patient data were collected from electronic medical records.

### 2.2. Evaluation of Response to Immunotherapy by Radiological Assessment

Radiological assessments were conducted before starting anti-PD-(L)1 treatment and every 10–12 weeks thereafter. Additional scans were performed at the physician’s discretion when deemed necessary. Patients were classified according to their response to the first radiological evaluation (10–12 weeks) into three groups: progressive disease (PD), stable disease (SD), and responders (R). Responders were further subclassified into partial response (PR) and complete response (CR). Response evaluation was assessed locally according to RECIST 1.1 or iRECIST criteria. Cases in which radiological evaluation could not be performed due to the patient’s death were classified as PD.

### 2.3. Immunophenotyping

One hundred microliters of whole blood were stained with anti-CD3-PECy7 (BioLegend, San Diego, CA, USA; clone HIT3a), anti-CD8-PECy5 (BioLegend; clone SK1), anti-PD-L1-PE (BioLegend; clone 29E.2A3), anti-CD14-PECy7 (BD Bioscience, San Jose, CA, USA; clone M5E2), anti-CD41a-FITC (Immunotools, Friesoythe, Germany; clone HIP8), and anti-CD62P-APC (Immunotools; clone HI62P). The cells were then incubated for 15 min in the dark before adding 2 mL of BD FACS lysing solution 1X (BD Bioscience) for 10 min. Finally, the cells were washed with 2 mL of PBS 1X and resuspended in 400 µL of PBS 1X before acquisition by flow cytometry (MACSQuant Analyzer 10 flow cytometer; Miltenyi Biotec, Bergisch Galdbach, Germany). Negative gates for each marker were defined by fluorescence minus one (FMO) controls.

### 2.4. Flow Cytometry Analysis

After doublet exclusion, lymphocytes were first gated based on their morphology by Side Scatter (SSC) and Forward Scatter (FSC). Specifically, CD4^+^ and CD8^+^ T lymphocytes were identified as CD3^+^ CD8^−^ and CD3^+^ CD8^+^ cells, respectively, in the lymphocytes gate. NK cells were gated as CD3^−^ CD8^+^ cells. Monocytes were selected according to positive CD14 expression and SSC, whereas neutrophils were identified by high SSC and negative CD14 expression. PLTs were identified as CD41a^+^ events in FSC and SSC plots on a logarithmic scale and activated PLTs as CD41a^+^ CD62P^+^ events. The PLT gate region was previously established using a blend of size-calibrated fluorescent beads with sizes ranging from 0.22 to 1.35 µm (Spherotech, Green Oaks, IL, USA). Leukocytes with bound PLTs were identified by previous specific markers plus CD41a^+^ cells, as shown in Appendix A. PD-L1 expression was analyzed in all leukocyte populations studied. The percentage of PD-L1 positive cells (PD-L1^+^), defined by a fluorescent minus one strategy for each population, was obtained using FlowJo version X (FlowJo LLC, Ashland, OR, USA).

During treatment, changes in the percentages of PD-L1^+^ leukocytes were categorized into three groups: those with a >10% decrease, those with no significant change (with less than a 10% change), and those with a >10% increase [25].

### 2.5. Determination of Soluble Mediators

The concentrations of sPD-L1 (Invitrogen, Carlsbad, CA, USA), IL-17A (Peprotech, London, UK), IL-6, IL-10 (Immunotools), IFN-γ (Mabtech, Nacka Strand, Sweden), and sIL-6R (BD Biosciences, San Jose, CA, USA), were measured in the plasma of NSCLC patients treated with PD-(L)1 at baseline and upon 10–12 weeks of follow-up (coinciding with the timing of the first radiological evaluation). All these soluble molecules were quantified using the standard curves provided by the corresponding ELISA kits [26]. The lower limits of detection were 4.69 pg/mL for sPD-L1, 10 pg/mL for IL-17A, 6.1 pg/mL for IL-6, 16 pg/mL for IL-10, 4 pg/mL for IFN-γ, and 2 pg/mL for sIL-6R.

### 2.6. Statistical Analysis

The statistical analysis was performed using GraphPad Prism Version 7 (GraphPad Software, La Jolla, CA, USA) and XLSTAT version 2021.4 software (Addinsoft, Paris, France). Normal data distribution was assessed by the Kolmogorov–Smirnov test. Qualitative variables were presented as numbers and percentages ± standard error of the mean (SEM) whereas quantitative variables were described as median plus interquartile range (IQR). The analysis of covariance (ANCOVA) method was used to examine the influence of smoking status on PD-L1^+^ leukocyte populations in NSCLC patients classified according to their response to anti-PD-(L)1 therapy. Comparisons between more than two groups were performed using the one-way analysis of variance (ANOVA) with the Bonferroni correction (parametric data) and the Kruskal–Wallis test with Dunn’s post hoc correction (non-parametric data). Paired variables, monitored every 4–6 weeks upon six months, were analyzed by ANOVA with the Tukey post hoc correction for parametric data and the Friedman test with Dunn’s post hoc correction for non-parametric data. A multinomial logistic regression analysis was performed to identify the risk factors associated with anti-PD-(L)1 response. Otherwise, the early effects of anti-PD-(L)1 therapy were assessed after 4–6 weeks of treatment using the χ^2^ test to compare changes among PD, SD, and R patients. All *p*-values were based on a two-sided hypothesis, and those under <0.05 were considered statistically significant. In regard to soluble factors, the Wilcoxon test was used to compare the baseline and 10–12 week time points.

Network analyses were performed to visualize complex and simultaneous interactions between PD-L1^+^ leukocytes and studied soluble mediators. This analysis was carried out 10–12 weeks after initiation for each of the three studied groups using the JASP 0.17.2 (Amsterdam, The Netherlands). This kind of analysis is grounded in a multivariate statistical approach, where variables are referred to as nodes and connections between the nodes are called edges [27]. Edges of a network can be interpreted as the full conditional association between two nodes after conditioning on all other nodes in the network. Blue edges indicate positive correlations, whereas red edges indicate negative correlations. Thicker and denser colored lines represent stronger relationships. To assess the importance of nodes in the network common centrality indices including strength, betweenness and closeness were reported. Strength indicates how strongly a node is directly associated with other nodes in the network. Betweenness measures the importance of a node in the average pathway between another pair of nodes. Closeness quantifies the indirect connectivity between a given node and all other nodes in the network.

## 3. Results

### 3.1. Patients’ Clinicopathological Characteristics

Patients’ clinicopathological characteristics are summarized in Table 1. Male patients constituted 77% of the cohort. The median age of the study participants was 65 years (36–84) with the majority being current or former smokers (90.3%). Non-squamous histology was the most frequent (71%). Baseline Eastern Cooperative Oncology Group Performance Status (ECOG PS) was 0 in 13.3%, 1 in 73.4%, and 2 in 13.3% of patients.

Tumor PD-L1 expression was evaluated by immunohistochemical staining with the 22C3 antibody (provided in the Dako PharmDx PD-L1 kit) in 85% of patients. Of these, PD-L1 was negative (<1%) in 28.1% of patients, low (1–49%) in 33.3%, and high (≥50%) in 38.6%. Anti-PD-(L)1 treatment was classified as first-line therapy in 32.7% of patients, second-line therapy in 53.1%, and third-line therapy or beyond in 14.2%. Most patients received PD-(L)1 inhibitors as monotherapy (86.8%). Anti-PD-1 was administrated to 75.2% of patients and anti-PD-L1 to 24.8%.

Radiological response after 10–12 weeks of treatment was PD in 48.7% of patients, SD in 21.2%, PR in 26.5%, and CR in 3.6% of patients. The response rate was 30.1%, and the disease control rate was 51.3%. As for the data cut-off in November 2022, 24 patients were still receiving treatment with PD-(L)1 inhibitors. We observed that the distribution by sex, age, and smoking status was comparable in all three groups of patients. However, the chi-square test revealed only significant differences in the frequencies of patients classified according to histology tumor types (χ^2^ = 11.7; *p* = 0.003), with the majority of R patients being squamous, whereas non-squamous histology was predominant in SD and PD patients. Further analysis revealed that tumor PD-L1 expression was not significantly associated with anti-PD-(L)1 therapy response.

### 3.2. Pretreatment PD-L1 Expression in Leukocytes and Platelets

Both R and SD patients showed higher percentages of circulating PD-L1^+^ CD14^+^ cells prior to the first administration of anti-PD-(L)1 compared to PD patients (Figure 1A). R patients had higher percentages of PD-L1^+^ neutrophils compared to PD patients (Figure 1B). Notably, no statistical differences were observed in PD-L1^+^ NK cells according to response groups (Figure 1C).

With regard to lymphoid lineage, PD patients exhibited higher percentages of PD-L1^+^ CD4^+^ T cells compared to SD patients (Figure 1D) and R patients showed higher percentages of PD-L1^+^ CD8^+^ T cells (Figure 1E). The percentages of PD-L1^+^ PLTs were significantly higher in R patients compared to PD patients (Figure 1F). We also observed that percentages of PD-L1^+^ CD14^+^ cells correlated positively with percentages of PD-L1^+^ neutrophils and PD-L1^+^ PLTs (Figure 1G,H). Nevertheless, PD-L1^+^ neutrophils did not correlate with PD-L1^+^ PLTs (Figure 1I). We did not find any correlation between PD-L1^+^ leukocyte subpopulations and PD-L1 tumor expression. Other PD-L1 sources, particularly sPD-L1, did not present significant differences according to response groups (Figure 1J). The percentages of circulating PD-L1^+^ leukocytes and platelets from HD are available in our previous article [20]. We further compared PD-L1 expression among the leukocyte populations according to tumor histology, and we found that there were no differences.

We verified by the ANCOVA method that PD-L1 expression on leukocyte populations in NSCLC patients classified according to their response to anti-PD-(L)1 therapy was not influenced by smoking status.

### 3.3. Progressive Changes in PD-L1 Expression on Leukocytes and Platelets

PD-L1 expression on distinct leukocyte subpopulations was monitored every 4–6 weeks upon six months of anti-PD-(L)1 therapy. Our investigation revealed a significant decline in the percentages of PD-L1^+^ CD14^+^ cells and PD-L1^+^ neutrophils within the initial 4–6 weeks of therapy in R patients but not in the other groups of patients. Particularly noteworthy, we also found a significant decrease in the percentage of PD-L1^+^ CD14^+^ cells between baseline and weeks 22–24 of treatment in R patients (Figure 2A,B).

Interestingly, the percentage of circulating PD-L1^+^ PLTs also tended to decrease within the initial 4–6 weeks of therapy in R patients (Figure 2C). Regarding NK cells, CD4^+,^ and CD8^+^ T cells, no substantial alterations were observed in PD-L1 expression during this period of time in any of the studied groups (Appendix A). These observations suggested the pre-existence of a threshold in PD-L1 expression needs to be surpassed for treatment effectiveness.

Significant differences between frequencies of patients were observed within the first 4–6 weeks of anti-PD-(L)1 therapy between the response groups. Notably, an increase >10% in the proportion of PD-L1^+^ CD14^+^ cells, PD-L1^+^ neutrophils, and PD-L1^+^ PLTs was more frequently observed in the PD group: 43.2% showed an increase in the proportion of PD-L1^+^ CD14^+^ cells, whereas this percentage did not exceed 18.2% for the other groups (Figure 2D); 47.7% displayed an increase in PD-L1^+^ neutrophils, whereas this proportion did not exceed 27.3% for the other groups (Figure 2E); and 43.8% exhibited an increase in PD-L1^+^ PLTs, whereas this did not exceed 28% for R patients (Figure 2F). Additionally, the SD group had the highest proportion of individuals with a decrease of >10% in the percentages of PD-L1^+^ NK cells, followed by R and PD patients (Appendix A). No differences were observed in the percentage of PD-L1^+^ CD4^+^ and CD8^+^ T cells between the response groups during this period (Appendix A).

### 3.4. Kinetics of PD-L1^+^ CD14^+^ Cells and Neutrophils with or without Bound Platelets

Next, we analyzed three possible scenarios to explain differences in PD-L1 expression on CD14^+^ cells and neutrophils observed in our cohort of patients. First, considering PLTs as a potential source of PD-L1, it is plausible to suppose that patients exhibited differences in the percentages of leukocyte-PLT complexes according to their response to anti-PD-(L)1. However, no significant differences neither before starting anti-PD-(L)1 therapy nor during 24 weeks of follow-up were observed between groups (Figure 3A–D).

We then compared PD-L1 expression on leukocytes with or without bound PLTs to determine the proportion of PD-L1 originating from leukocyte intrinsic expression. The percentage of PD-L1^+^ CD14^+^ cells and PD-L1^+^ neutrophils without bound PLTs decreased after 22–24 weeks of treatment compared to a 4–6-week time point only in PD patients (Figure 3E,F). We also observed that the pretreatment percentage of PD-L1^+^ CD14^+^ cells, with and without bound PLTs, was significantly lower in PD patients than in SD and R patients (Figure 3G).

### 3.5. Determination of Plasma Soluble Immunomodulatory Factors’ Concentrations and Network Analysis

We observed significant differences in plasma IL-17 concentrations between SD and R patients at baseline (Figure 4A). However, no significant differences were observed in plasma IFN-γ, IL-10, or IL-6, nor in IL-6R concentrations (Figure 4B–E). Notably, we observed that plasma IL-17 concentration increased in R patients after 10–12 weeks of anti-PD-(L)1 treatment but tended to decrease in SD patients (Figure 4A).

With all these results, we computed a Network Analysis, not restricted to significant variables, to visualize the potential interactions between PD-L1^+^ leukocyte populations and plasma cytokines in NSCLC patients classified according to their response to anti-PD-(L)1 treatment (Figure 5A). Different connectivity patterns were observed for all groups of patients after 10–12 weeks of anti-PD-(L)1 treatment: (1) in both PD and SD patients, sIL-6R (node 9) was negatively associated with PD-L1^+^ CD14^+^ cells (node 4), whereas this molecule was positively associated with PD-L1^+^ NK cells (node 3) in R patients; (2) a strong connectivity between PD-L1^+^ myeloid and lymphoid cells in R patients but not in PD and SD patients; (3) a strong positive correlation between PD-L1^+^ neutrophils (node 5) and IL-10 (node 8) and between IL-17 (node 6) and IFN-γ (node 7) in PD patients which was not observed in SD and R patients.

The centrality indices of these network analyses are represented in Figure 5B. After 10–12 weeks of anti-PD-(L)1 treatment, the percentage of PD-L1^+^ CD8^+^ T cells exhibited the highest betweenness and strength in R patients. Moreover, plasma sIL-6R concentrations presented higher betweenness in R and SD patients compared to PD patients. This molecule also displayed the highest strength in SD. The percentage of PD-L1^+^ CD14^+^ cells had higher betweenness and closeness in PD and SD patients compared to R.

### 3.6. Multinomial Logistic Regression Model to Predict Response to Anti-PD-(L)1

To assess the contribution to predicting the response to anti-PD-L1 therapy of those variable that significantly differ at baseline among the studied groups, we conducted a multivariate multinomial logistic regression model, with R patients considered as the reference category (Figure 6A). The model fitting information (Appendix A) showed that the full model predicts response to anti-PD-L1 treatment significantly better than the intercept-only model alone. Compared to SD and PD patients, a higher percentage of PD-L1^+^ CD14^+^ cells PLT^neg^ in an NSCLC patient was considered a protective factor, significantly increasing the probability of responding to anti-PD-(L)1 therapy. However, higher IL-17 concentrations were considered a risk factor, decreasing the probability of responding to anti-PD-(L)1 therapy. Additionally, compared to PD, the percentage of PD-L1^+^ neutrophils was also identified as a protective factor. Notably, the pretreatment PD-L1^+^ CD14^+^ cells/IL-17 ratio was significantly higher in R patients compared to PD, and tended to be higher compared to SD (Figure 6B). We did not observe significant differences in PD-L1^+^ CD14^+^ cells/IL-17 ratios after 10–12 weeks of anti-PD-(L)1 treatment among studied groups. 

## 4. Discussion

Our study highlights the presence of higher percentages of PD-L1^+^ CD14^+^ cells, PD-L1^+^ neutrophils, and PD-L1^+^ PLTs before anti-PD-(L)1 therapy initiation in R patients, but not in SD and PD patients. This finding suggests the pre-existence of an immune milieu associated with the expression of PD-L1 that plays a crucial role in determining patient response. Additionally, this observation may point to the presence of a threshold in the proportion of PD-L1^+^ leukocytes that needs to be surpassed for the treatment to be effective. Consequently, in patients with a low percentage of PD-L1^+^ leukocytes (PD patients), the administration of anti-PD-(L)1 treatment may confer no therapeutic benefit as the tumor may be employing alternative immune evasion mechanisms such as TIM-3/Galectin-9 [28] or TIGIT/CD155 [29,30]. These findings are in line with those we reported previously, where we associated these three populations with longer PFS in NSCLC patients [20]. Here, further exploration of the kinetics of percentages of PD-L1^+^ leukocytes and PLTs unveiled significant changes in PD-L1 expression on CD14^+^ cells, neutrophils, and PLTs within the first 4–6 weeks of treatment, indicating that the treatment has a rapid effect and that it is not necessary to monitor these populations for up to 6 months since no significant changes will be observed. This confers clinical advantages in promptly identifying non-responder patients and allowing for the exploration of further treatments while reducing healthcare costs and toxicity.

Information concerning the relevance of monocytes in predicting the response to PD-(L)1 inhibitors is controversial. On the one hand, in NSCLC patients, higher absolute monocyte counts before starting PD-(L)1 inhibitors correlated with a shorter time to response [31]. Moreover, Krieg et al. proposed that abundant circulating CD14^+^ CD16^−^ HLA-DR^hi^ cells before anti-PD-1 therapy facilitate the infiltration of circulating T lymphocytes into the tumor, enhancing the cytotoxicity against cancer cells [32]. Earlier studies also reported that HLA-DR expression on tumor-infiltrating monocytes correlates with high levels of PD-L1 [33], suggesting that the increased PD-L1^+^ CD14^+^ cells found in R patients in our cohort may have these functional characteristics. On the other hand, PD-L1 expression on circulating monocytes has been associated with shorter survival in patients treated with anti-PD-1 therapy [34,35]. Apparent discrepancies may be attributed to the heterogeneity of the cohorts, PD-1 therapy, tumor type, and stage. Moreover, some studies analyzed levels of expression of PD-L1 whereas we analyzed percentages of PD-L1^+^ cells.

Neutrophils have largely been limited to the measurement of the blood neutrophil-to-lymphocyte ratio and an elevated ratio is associated with shorter OS and PFS in NSCLC patients treated with PD-(L)1 inhibitors [36]. However, in line with our findings, recent studies have focused on a subset of circulating low-density neutrophils with higher levels of PD-L1 (transcriptomic data) and mature phenotypes that mediate immunosuppression and may be the target of PD-(L)1 inhibitors [37]. In other types of cancer, results were more variable; in the case of melanoma patients, lower frequencies of PD-L1^+^ neutrophils were associated with better PFS [38]. Differences in the findings of this study with ours could be explained by the different tumor types, tumor stages, and types of PD-(L)1 inhibitor therapy. Further experiments need to be performed to better characterize PD-L1^+^ neutrophils phenotypically and functionally in our cohort.

There has been growing interest in exploring the potential predictive value of PD-L1 expression on PLTs in the context of response to PD-(L)1 inhibitors. In line with our findings, Colarusso et al. demonstrated that increased PD-L1 expression in PLTs correlated with a longer clinical benefit from anti-PD-L1 therapy in NSCLC patients [39]. Some studies have pointed to the transfer of PD-L1 protein from tumor cells to blood PLTs during their interaction in a fibronectin- and integrin α5β1-dependent manner [24]. However, we did not find any correlation between PD-L1 expression by tumor cells and by PLTs. Zaslavsky et al. documented PD-L1 expression on circulating PLTs even in patients with PD-L1 negative tumors, further highlighting the complexity of PD-L1 dynamics within the tumor microenvironment [40]. Possible explanations for this contradiction may include (1) limitations in the measurement of PD-L1 expression in tumor biopsies, including sample quality and tumor heterogeneity; (2) PD-L1 expressed by PLTs may originate from their own expression and the transference from tumor cells. Interestingly, our findings reveal that PLTs are a major contributor to PD-L1 expression on leukocytes, irrespective of patients’ response, as the percentages of PD-L1^+^ leukocytes were higher when PLTs were bounded. Additionally, significant changes in the frequencies of PD-L1^+^ leukocytes were limited to populations with bound PLTs, and not in those without bound PLTs. This observation, combined with a similar kinetic profile of PD-L1^+^ PLTs and PD-L1^+^ myeloid cells, suggests that PD-(L)1 inhibitors specifically target PD-L1 derived from PLTs, even when they are bound to leukocytes, rather than PD-L1 from leukocytes themselves.

Lower plasma IL-17 concentration and its significant increase in responders, together with the identification of IL-17 as a risk factor in non-responders to anti-PD-(L)1 therapy, can help to understand the response to PD-(L)1 inhibitors. Notably, at baseline, R patients have lower IL-17 levels compared to SD patients but only show a trend toward lower levels than PD patients (*p* = 0.1). One plausible explanation for these findings is the line of ICI treatment. SD and R patients mainly undergo first- and second-line treatments, whereas PD patients predominantly receive second-line treatment. These treatment line variations may not only contribute to baseline differences but also influence the direction of IL-17 level changes (i.e., decrease/increase) within each group. Further investigations with a larger patient cohort are imperative to understand the impact of treatment lines. Our network analysis also revealed an association of IFN-γ with IL-17 in PD patients but not in R patients. We additionally observed a positive association between IL-10 and PD-L1^+^ neutrophils in PD patients only. In line with these results, Zhang et al. reported that tumor-associated neutrophils secrete IL-10 to promote tumor progression and distant metastasis in lung cancer [41]. The negative association between IL-6R and the percentage of PD-L1^+^ CD14^+^ cells in PD and SD, but not in R patients, may be explained by the predominance of the IL-6 trans-signaling pathway. It is described that in cancer conditions, IL-6 enhances PD-L1 expression in monocytes [42] and IL-6 trans-signaling limits the classical pathway, resulting in increased angiogenesis, tumor growth, and metastasis [43]. Despite that we did not find differences in IL-6 among NSCLC patients classified according to anti-PD-(L)1 response, in other cancers, such as melanoma and breast cancer, IL-6 levels at baseline have been correlated with rapid disease progression [44,45].

Despite the relevance of our findings, our study has some limitations. First, we lost some patients during the follow-up, particularly among those who progressed to PD-(L)1 inhibitors, who discontinued treatment, experienced adverse effects, or passed away. Second, measuring PD-L1 on CD14^+^ cells or neutrophils alone is insufficient to determine the effectiveness of anti-PD-(L)1 treatment, since other factors such as tumor mutations and immune cell infiltration may also play significant roles in predicting treatment response. Third, the NSCLC cohort of patients included in our study was heterogeneous, also including patients receiving anti-PD-(L)1 treatment in combination with chemotherapy or other immune checkpoint inhibitors and in different treatment lines, which may have influenced our results. Extended research is needed to assess whether these results can be extrapolated to other lung cancer subtypes. Nevertheless, our findings suggest that PD-L1 expression on circulating cells, in addition to PD-L1 expression in tumors, is relevant to determining the response to PD-(L)1 inhibitors.

Although further investigation is necessary to fully elucidate their precise implications, our findings suggest the critical role of pre-existing immune landscapes and treatment-induced changes in PD-L1 expression on specific leukocyte populations. These factors could provide one of the mechanistic foundations for patient stratification and timely therapeutic decisions, ultimately enhancing the efficacy and cost-effectiveness of PD-(L)1 inhibitor treatments.

## Figures and Tables

**Figure 1 biomedicines-12-00958-f001:**
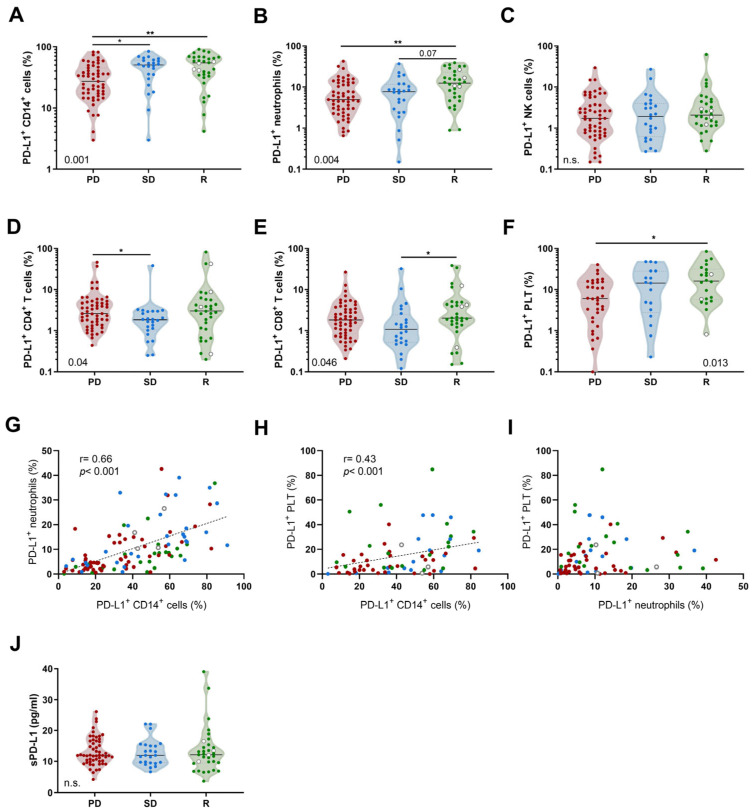
Pretreatment percentages of PD-L1^+^ leukocytes and platelets in NSCLC patients based on anti-PD-(L)1 response. Comparison of PD-L1^+^ (**A**) CD14^+^ cells, (**B**) neutrophils, (**C**) NK cells, (**D**) CD4^+^ T cells, (**E**) CD8^+^ T cells, and (**F**) platelets among PD (n = 55; except for platelets, n = 34), SD (n = 24; except for platelets, n = 17) and R patients (n = 34; except for platelets, n = 22). (**G**–**I**) Spearman correlations between PD-L1^+^ CD14^+^ cells, PD-L1^+^ neutrophils, and PD-L1^+^ platelets. (**J**) Comparison of the plasma sPD-L1 concentrations between PD, SD, and R patients. The white dots represent patients with complete response and green dots represent patients with partial response. Statistical analysis between three groups of patients was performed using the Kruskal–Wallis test with Dunn’s correction. *p*-values are shown in graphs (*) *p* < 0.05; (**) *p* < 0.01. n.s. is not significant.

**Figure 2 biomedicines-12-00958-f002:**
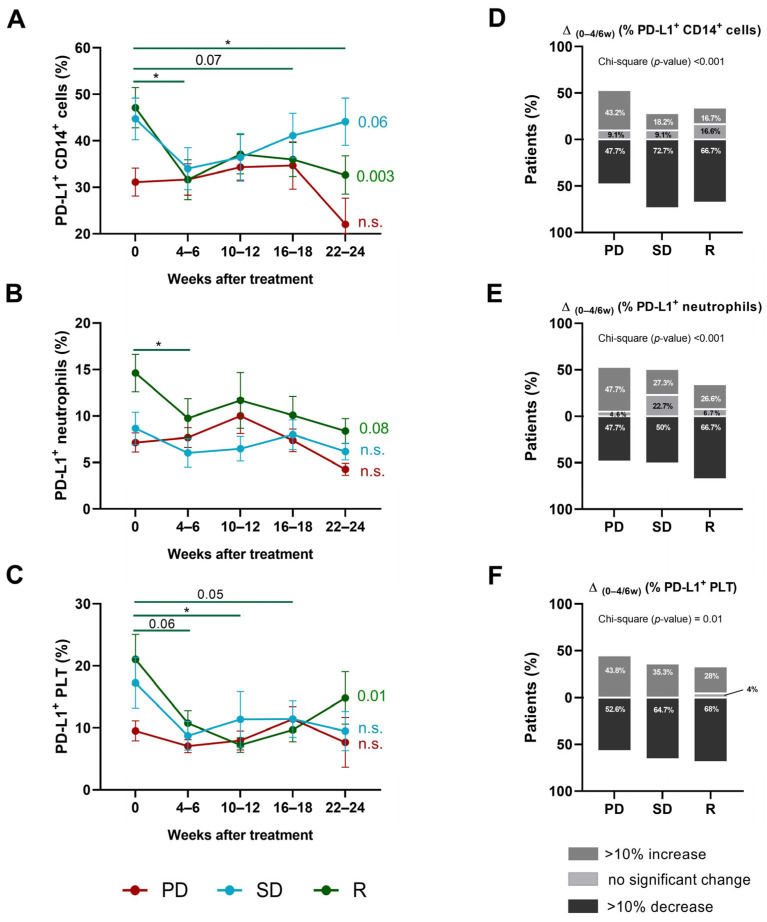
Kinetics of PD-L1^+^ CD14^+^ cells, PD-L1^+^ neutrophils, and PD-L1^+^ platelets in NSCLC patients based on anti-PD-(L)1 response. Percentages of (**A**) PD-L1^+^ CD14^+^ cells, (**B**) PD-L1^+^ neutrophils, and (**C**) PD-L1^+^ platelets during 24 weeks of anti-PD-(L)1 treatment in PD (red line), SD (blue line), and R patients (green line). *p*-values from one-way ANOVAs for each of the three groups are displayed on the right side of each graph, and significant differences between successive time intervals are analyzed by Tukey post hoc comparisons and marked with a line. Changes (>10% decrease (black), no significant change (with less than a 10% change) (light grey), and >10% increase (dark grey)) in the percentages of (**D**) CD14^+^ cells, (**E**) neutrophils, and (**F**) platelets in the first 4–6 weeks of anti-PD-(L)1 treatment. (*) *p* < 0.05; n.s. is not significant.

**Figure 3 biomedicines-12-00958-f003:**
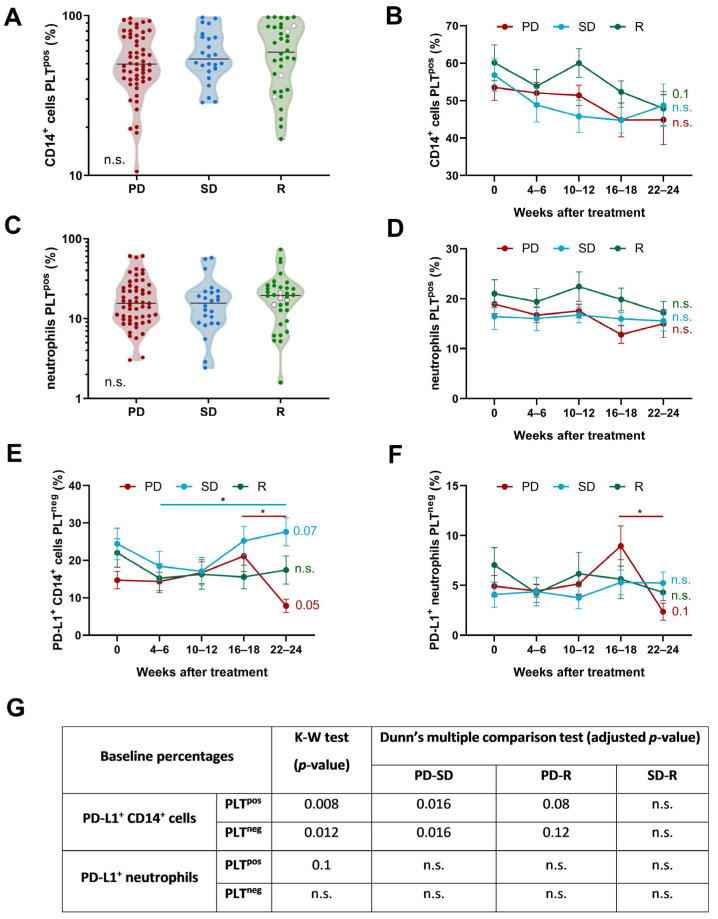
Kinetics of CD14^+^ cells and neutrophils with or without bound platelets and their PD-L1 expression in NSCLC patients based on anti-PD-(L)1 response. (**A**,**B**) CD14^+^ cells and (**C**,**D**) neutrophils with bound platelets and their evolution throughout 22–24 weeks of treatment in PD (n = 55), SD (n = 24), and R patients (n = 34). White dots represent patients with complete response and green dots patients with partial response. Kinetics of PD-L1^+^ (**E**) CD14^+^ cells and (**F**) neutrophils without bound platelets during 24 weeks of anti-PD-(L)1 therapy. *p*-values from one-way ANOVAs for each of the three groups are displayed on the right side of each graph, and significant differences between successive time intervals are analyzed by the Tukey post hoc comparisons and marked with a line. (**G**) Comparison of pretreatment baseline percentages of PD-L1^+^ CD14^+^ cells and PD-L1^+^ neutrophils with or without bound platelets among NSCLC classified according to their response to anti-PD-(L)1 therapy. Statistical analyses between three groups of patients were performed using Kruskal–Wallis test with Dunn’s correction (showed in the table). (*) *p* < 0.05; n.s. is not significant.

**Figure 4 biomedicines-12-00958-f004:**
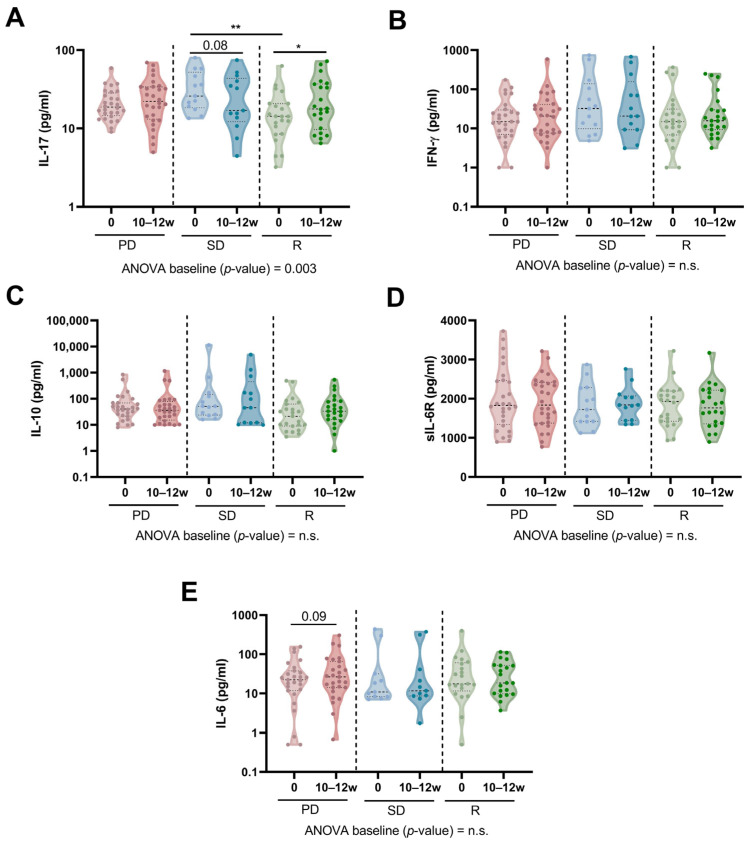
Concentration of plasma-soluble immunomodulatory factors in NSCLC patients based on anti-PD-(L)1 response. (**A**) IL-17, (**B**) IFN-γ, (**C**) IL-10, (**D**) sIL-6R, and (**E**) IL-6 concentrations (pg/mL) in plasma from PD, SD, and R patients measured at baseline and after 10–12 weeks of treatment. *p*-values are shown in graphs (*) *p* < 0.05; (**) *p* < 0.01; n.s. is not significant. The Kruskal–Wallis test with Dunn’s correction was used to compare baseline concentrations between studied groups, whereas the Wilcoxon test was used to compare baseline and 10–12 week time points for each soluble factor.

**Figure 5 biomedicines-12-00958-f005:**
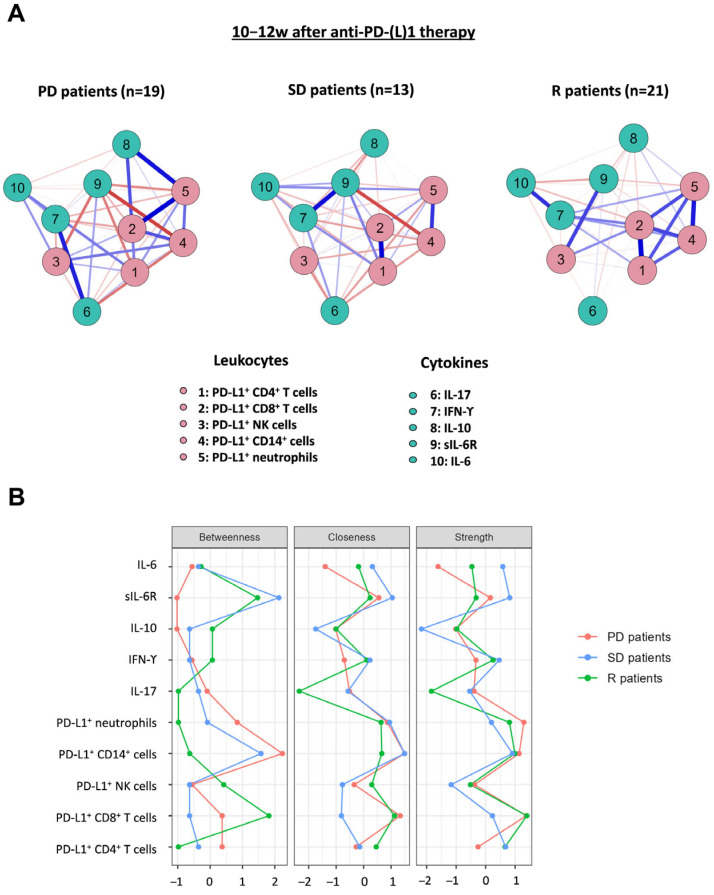
Network diagrams and centrality plots for NSCLC patients treated with anti-PD-(L)1-response at baseline and after 10–12 weeks of treatment. (**A**) Network model of leukocytes and cytokines interactions in PD (n = 19), SD (n = 13), and R (n = 21) NSCLC after 10–12 weeks of anti-PD-(L)1 therapy. Blue lines represent positive associations, and red lines negative associations. Thicker lines indicate stronger associations. Red nodes represent PD-L1^+^ leukocyte populations and the green ones represent plasma cytokines. Network diagrams were estimated by correlation method. (**B**) Centrality indices for the network nodes. The *x*-axis is the raw values of centrality indices and the *y*-axis represents all the nodes.

**Figure 6 biomedicines-12-00958-f006:**
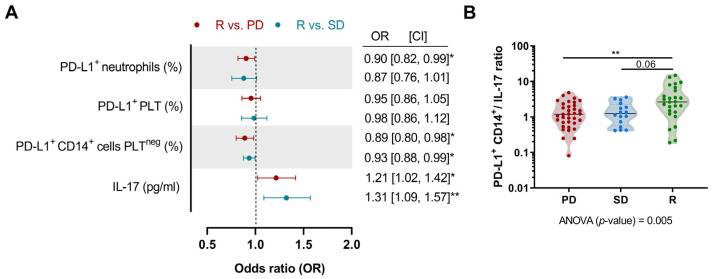
Multinomial logistic regression analysis and pretreatment PD-L1^+^ CD14^+^ cells/IL-17 ratio. (**A**) The reference group is R patients (PD vs. R in red; SD vs. R in blue). Dots represent the odds ratio and the error bars represent the 95% CI. (**B**) Ratio between the percentage of PD-L1^+^ CD14^+^ cells and plasma IL-17 concentration before starting anti-PD-(L)1 therapy. The Kruskal–Wallis test with Dunn’s correction was used to compare the baseline ratio between studied groups. *p*-values are shown in graphs (*) *p* < 0.05; (**) *p* < 0.01. CI, confidence interval.

**Table 1 biomedicines-12-00958-t001:** Baseline characteristics of NSCLC patients.

Characteristic	Patients (n = 113)	R Patients (n = 34)	SD Patients (n = 24)	PD Patients (n = 55)
**Sex, No. (%)**				
Male	87 (77%)	28 (82.4%)	14 (58.3%)	45 (81.8%)
Female	26 (23%)	6 (17.6%)	10 (41.7%)	10 (18.2%)
**Age, years** Median (range)	65 (36–84)	65 (45–83)	64 (35–86)	66 (48–85)
**Smoking status**				
Current smokers	34 (30.1%)	13 (38.2%)	7 (29.2%)	14 (25.5%)
Former smokers	68 (60.2%)	19 (55.9%)	14 (58.3%)	35 (63.6%)
Never smokers	11 (9.7%)	2 (5.9%)	3 (12.5%)	6 (10.9%)
**Histology**				
Squamous	42 (37.2%)	20 (58.8%)	6 (25%)	16 (29.1%)
Non-Squamous	71 (62.8%)	14 (41.2%)	18 (75%)	39 (70.9%)
**PD-L1 tumor expression ***				
Positive	69 (71.9%)	22 (71%)	16 (76.2%)	31 (70.5%)
Low: <1–49%	35	10	7	15
High: ≥50%	34	12	9	16
Negative	27 (28.1%)	9 (29%)	5 (23.8%)	13 (29.5%)
**Treatment Line**				
First Line	37 (32.7%)	16 (47.1%)	10 (41.7%)	11 (20%)
Second Line	60 (53.1%)	13 (38.2%)	12 (50%)	35 (63.6%)
Third Line and Beyond	16 (14.2%)	5 (14.7%)	2 (8.3%)	9 (16.4%)
**Drug**				
Anti-PD-L1	28 (24.8%)	4 (11.8%)	9 (37.5%)	15 (27.3%)
Anti-PD-1	85 (75.2%)	30 (88.2%)	15 (62.5%)	40 (72.7%)
**Radiological Response**				
Complete Response (CR)	4 (3.6%)	4 (11.8%)	-	-
Partial Response (PR)	30 (26.5%)	30 (88.2%)	-	-
Stable Disease (SD)	24 (21.2%)	-	24 (100%)	-
Progression Disease (PD)	55 (48.7%)	-	-	55 (100%)

* PD-L1 tumor expression was only measured in 96 patients, and the percentages were calculated based on these patients. R, responders; SD, stable disease; PD, progressors. We conducted a chi-square test to compare frequencies, and we have only observed significant differences regarding the histology tumor type (*p* = 0.003).

## Data Availability

The data presented in this study are available from corresponding authors.

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
