# Peer review of "Potential Role of Circulating PD-L1+ Leukocytes as a Predictor of Response to Anti-PD-(L)1 Therapy in NSCLC Patients"

_biomedicines, 2024, doi:10.3390/biomedicines12050958_

Round 1

Reviewer 1 Report

Comments and Suggestions for Authors

the work is potentially interesting

Changes are necessary

1. in the introductory part regarding PD-1 immunotherapy, write that it is also used for other tumors such as melanoma and cite the paper: PMID: 37806113

2. replace the name Whole blood staining in the material and method section with the name immunophenotype

3. at the end of the chapter ELISA measurement of soluble mediators, add the reference as recommended in the literature: PMID: 33131355

4. in the discussion, he added that among the many cytokines after PDl therapy, it was shown that IL-6 is a negative regulator of the response to therapy, as shown in the paper: PMID: 36754615

Comments on the Quality of English Language

ok

Reviewer 2 Report

Comments and Suggestions for Authors

Major comments:

1.     Please have a table or shown in Table 1 to compare the characteristics among PD, SD, and R, including smoking status, histology, tumor stage, treatment line, and so one since other characteristics may also contribute the radiological outcomes.

2.     Do tumor PD-L1 levels predict the radiologic responses in the enrolled patients?

3.     Do tumor PD-L1 correlate with PD-L1+CD14+, PD-L1+neutrophils, and PD-L1+PLT?

4.     Although PD-L1+CD14+, PD-L1+neutrophils, and PD-L1+PLT increase in R group, but these three indicators are not increased after anti-PD-(L)1 treatment (Fig 2). Meanwhile, the levels become decrease and similar among the three groups after initial 4-6W. It is hard to convincing that these three indicators contribute to the different outcomes as reliable biomarkers.

5.     Why suddenly compare 16-18W with 22-24W in Fig 3E and Fig 3F. One-way ANOVA should be used to clarify the statistical significance, rather than just compare the specific time points.

6.     Why just compare 10-12W in Fig 4. How about other groups?

7.     IL-17 decreases in SD group after 10-12W, but IL-17 increases in R group after 10-12W. Particularly, pre-treatment IL-17 levels in R group are similar to PD, both are lower than SD. It is hard to convincing that IL-17 is a reliable biomarker.

8.     The network diagrams should just use the indicators with statistical significance to predict the radiological responses.

9.     How about the PD-L1+CD14+/IL-17 ratio after anti-PD-(L)1 treatment, does which still predict the radiological responses.

10.  In the final conclusions, it is inappropriate that stat “rapid treatment-induced changes in PD-L1 expression on specific leukocyte populations, providing a foundation for precise patient stratification and timely therapeutic decisions”. In fact, these three indicators become decrease and similar among the three groups after initial 4-6W (Fig 2). The final figure (Fig 6B) just shows PD-L1+CD14+/IL-17 ratio before anti-PD-(L)1 treatment. How the rapid treatment-induced changes in PD-L1 expression provide timely therapeutic decisions?

Round 2

Reviewer 1 Report

Comments and Suggestions for Authors

accept

Comments on the Quality of English Language

minor

Reviewer 2 Report

Comments and Suggestions for Authors

No more questions.